

# An extensive comparison of species-abundance distribution models

Elita Baldridge[1,2], David J. Harris[3], Xiao Xiao[1,2,4,5] and Ethan P. White[1,2,3,6]

[1] Department of Biology, Utah State University, Logan, UT, United States
[2] Ecology Center, Utah State University, Logan, UT, United States
[3] Department of Wildlife Ecology and Conservation, University of Florida, Gainesville, FL, United States
[4] School of Biology and Ecology, University of Maine, Orono, ME, United States
[5] Senator George J. Mitchell Center for Sustainability Solutions, University of Maine, Orono, ME, United States
[6] Informatics Institute, University of Florida, Gainesville, FL, United States

## ABSTRACT

A number of different models have been proposed as descriptions of the species-abundance distribution (SAD). Most evaluations of these models use only one or two models, focus on only a single ecosystem or taxonomic group, or fail to use appropriate statistical methods. We use likelihood and AIC to compare the fit of four of the most widely used models to data on over 16,000 communities from a diverse array of taxonomic groups and ecosystems. Across all datasets combined the log-series, Poisson lognormal, and negative binomial all yield similar overall fits to the data. Therefore, when correcting for differences in the number of parameters the log-series generally provides the best fit to data. Within individual datasets some other distributions performed nearly as well as the log-series even after correcting for the number of parameters. The Zipf distribution is generally a poor characterization of the SAD.

## INTRODUCTION

The species abundance distribution (SAD) describes the full distribution of commonness and rarity in ecological systems. It is one of the most fundamental and ubiquitous patterns in ecology, and exhibits a consistent general form with many rare species and few abundant species occurring within a community. The SAD is one of the most widely studied patterns in ecology, leading to a proliferation of models that attempt to characterize the shape of the distribution and identify potential mechanisms for the pattern (see *McGill et al., 2007* for a recent review of SADs). These models range from arbitrary distributions that are chosen based on providing a good fit to the data (*Fisher, Corbet & Williams, 1943*), to distributions chosen based on the most likely states of generic random systems (*Frank, 2011*; *Harte, 2011*; *Locey & White, 2013*), to models based more directly on ecological processes (*Tokeshi, 1993*; *Hubbell, 2001*; *Volkov et al., 2003*; *Alroy, 2015*).

Which model or models provide the best fit to the data, and the resulting implications for the processes structuring ecological systems, is an active area of research (e.g., *McGill, 2003*; *Volkov et al., 2003*; *Ulrich, Ollik & Ugland, 2010*; *White, Thibault & Xiao, 2012*; *Connolly*

Corresponding author
Ethan P. White, ethan@weecology.org

*et al., 2014*). However, most comparisons of the different models: (1) use only a small subset of available models (typically two; e.g., *McGill, 2003*; *Volkov et al., 2003*; *White, Thibault & Xiao, 2012*; *Connolly et al., 2014*); (2) focus on a single ecosystem or taxonomic group (e.g., *McGill, 2003*; *Volkov et al., 2003*); or (3) fail to use the most appropriate statistical methods (e.g., *Ulrich, Ollik & Ugland, 2010*, see *Matthews & Whittaker, 2014* for discussion of best statistical methods for fitting SADs). This makes it difficult to draw general conclusions about which, if any, models provide the best empirical fit to species abundance distributions.

Here, we evaluate the performance of four of the most widely used models for the species abundance distribution using likelihood-based model selection on data from 16,209 communities and nine major taxonomic groups. This includes data from terrestrial, aquatic, and marine ecosystems representing roughly 50 million individual organisms in total.

## METHODS

### Data

We compiled data from citizen science projects, government surveys, and literature mining to produce a dataset with 16,209 communities, from nine taxonomic groups, representing nearly 50 million individual terrestrial, aquatic, and marine organisms. Data for trees, birds, butterflies and mammals was compiled by *White, Thibault & Xiao (2012)* from six data sources: the US Forest Service Forest Inventory and Analysis (FIA; *USDA Forest Service, 2010*), the North American Butterfly Association's North American Butterfly Count (NABC; *North American Butterfly Assoc, 2009*), the Mammal Community Database (MCDB; *Thibault et al., 2011*), Alwyn Gentry's Forest Transect Data Set (Gentry; *Phillips & Miller, 2002*), the Audubon Society Christmas Bird Count (CBC; *National Audubon Society, 2002*), and the US Geological Survey's North American Breeding Bird Survey (BBS; *Pardieck, Ziolkowski Jr & Hudson, 2014*) (see Table 1 for details). The publicly available datasets (FIA, MCDB, Gentry, and BBS) were acquired using the EcoData Retriever (http://data-retriever.org; *Morris & White, 2013*). Details of the treatment of these datasets can be found in Appendix A of *White, Thibault & Xiao (2012)*, but in general data were analyzed at the level of the site defined in the dataset and a single year of data was selected for each site. We modified the data slightly by removing sites 102 and 179 from the Gentry data due to issues with decimal abundances appearing in raw data due to either data entry or data structure errors. Data on Actinopterygii, Reptilia, Coleoptera, Arachnida, and Amphibia, were mined from literature by Baldridge and are publicly available (*Baldridge, 2013*) (see Table 1 for details). These data were collected at the level of the site defined in the publication if raw data were available at that scale, and at the scale of the entire study otherwise. The time scale of collection for this data depended on the study but was typically one or a few years. All data sources used in the analysis were samples (or censuses) of a taxonomic assemblage, where all individuals of any species observed are recorded. Abundances in the compiled datasets were counts of individuals.

**Table 1  Details of datasets used to evaluate the form of the species abundance distribution.** Datasets marked as private were obtained through data requests to the providers.

| Dataset | Dataset code | Availability | Total sites | Citation |
|---|---|---|---|---|
| Breeding bird survey | BBS | Public | 2,769 | *Pardieck, Ziolkowski Jr & Hudson (2014)* |
| Christmas bird count | CBC | Private | 1,999 | *National Audubon Society (2002)* |
| Gentry's forest transects | Gentry | Public | 220 | *Phillips & Miller (2002)* |
| Forest inventory and analysis | FIA | Public | 10,355 | *USDA Forest Service (2010)* |
| Mammal community database | MCDB | Public | 103 | *Thibault et al. (2011)* |
| NA butterfly count | NABA | Private | 400 | *North American Butterfly Assoc (2009)* |
| Actinopterygii | Actinopterygii | Public | 161 | *Baldridge (2013)* |
| Reptilia | Reptilia | Public | 129 | *Baldridge (2013)* |
| Amphibia | Amphibia | Public | 43 | *Baldridge (2013)* |
| Coleoptera | Coleoptera | Public | 5 | *Baldridge (2013)* |
| Arachnida | Arachnida | Public | 25 | *Baldridge (2013)* |

## Models

We selected models for analysis based on four criteria. First, since the majority of species abundance distributions (SADs) are constructed using counts of individuals (for discussion of alternative approaches see *McGill et al., 2007* and *Morlon et al., 2009*) we selected models with discrete distributions (i.e., those that only have non-zero probabilities for positive integer values of abundance). Second, in order to use best practices for comparing species abundance distributions we selected models with analytically defined probability mass functions that allow the calculation of likelihoods (see details in Analysis). Third, *McGill et al. (2007)* classified species abundance distribution models into five different families: purely statistical, branching process, population dynamics, niche partitioning, and spatial distribution of individuals. We evaluated models from each of these families, with some models having been derived from more than one family of processes. Finally, we selected models that have been widely used in the ecological literature. Based on these criteria we evaluated the log-series, the Poisson lognormal, the negative binomial, and the Zipf distributions. All distributions were defined to be capable of having non-zero probability at integer values from 1 to infinity.

The log-series is one of the first distributions used to describe the SAD, being derived as a purely statistical distribution by *Fisher, Corbet & Williams (1943)*. It has since been derived as the result of ecological processes, the metacommunity SAD for ecological neutral theory (*Hubbell, 2001*; *Volkov et al., 2003*), and several different maximum entropy models (*Pueyo, He & Zillio, 2007*; *Harte et al., 2008*).

The lognormal is one of the most commonly used distributions for describing the SAD (*McGill, 2003*) and has been derived as a null form of the distribution resulting from the central limit theorem (*May, 1975*), population dynamics (*Engen & Lande, 1996*), and niche partitioning (*Sugihara, 1980*). We use the Poisson lognormal because it is a discrete form of the distribution appropriate for fitting discrete abundance data (*Bulmer, 1974*).

The negative binomial (which can be derived as a Gamma-distributed mixture of Poisson distributions) provides a good characterization of the SAD predictions for several different
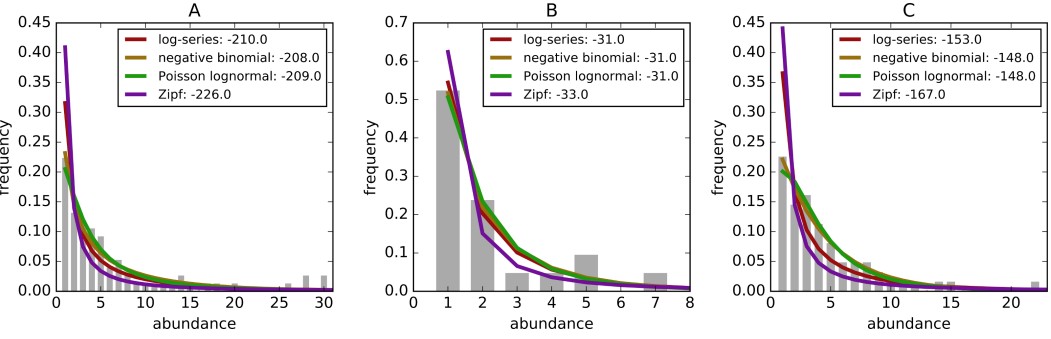

**Figure 1** Example species-abundance distributions including the empirical distributions (grey bars) and the best fitting log-series: maroon, negative binomial: brown, poisson lognormal: green, and Zipf: purple. Distributions are for (A) Breeding Bird Survey—Route 36 in New York, (B) Forest Inventory and Analysis—Unit 4, County 57, Plot 12 in Alabama, and (C) Gentry—Araracuara High Campina site in Colombia. Log-likelihoods of the models are included after the colon in the legend.

ecological neutral models for the purposes of model selection (*Connolly et al., 2014*). We use it to represent neutral models as a class.

The Zipf (or power law) distribution was derived based on both branching processes and as the outcome of the *McGill & Collin*'s (*2003*) spatial model. It was one of the best fitting distributions in a recent meta-analysis of SADs (*Ulrich, Ollik & Ugland, 2010*). We use the discrete form of the distribution which is appropriate for fitting discrete abundance data (*White, Enquist & Green, 2008*).

Figure 1 shows three example sites with the empirical distribution and associated models fit to the data. Zipf distributions tend to predict the most rare species followed by the log-series, the negative binomial, and Poisson lognormal.

## Analysis

Following current best practices for fitting distributions to data and evaluating their fit, we used maximum likelihood estimation to fit models to the data (*Clark, Cox & Laslett , 1999*; *Newman, 2005*; *White, Enquist & Green, 2008*) and likelihood-based model selection to compare the fits of the different models (*Burnham & Anderson, 2002*; *Edwards et al., 2007*). This approach has recently been affirmed as best practice for species abundance distributions (*Connolly et al., 2014*; *Matthews & Whittaker, 2014*). This requires that likelihoods for the models can be solved for and therefore we excluded models that lack probability mass functions and associated likelihoods. While methods have been proposed for comparing models without probability mass functions in this context (*Alroy, 2015*), these methods have not been evaluated to determine how well they perform compared to the widely accepted likelihood-based approaches.

For model comparison we used corrected Akaike Information Criterion (AICc) weights to compare the fits of models while correcting for differences in the number of parameters and appropriately handling the small sample sizes (i.e., numbers of species) in some communities (*Burnham & Anderson, 2002*). The Poisson lognormal and the negative binomial each have two fitted parameters, while the log-series and the Zipf distributions have one fitted parameter each. The model with the greatest AICc weight in each community

was considered to be the best fitting model for that community. We also assessed the full distribution of AICc weights to evaluate the similarity of the fits of the different models.

In addition to evaluating AICc of each model, we also examined the log-likelihood values of the models directly. We did this to assess the fit of the model while ignoring corrections for the number of parameters and the influence of similarities to other models in the set of candidate models. This also allows us to make more direct comparisons to previous analyses that have not corrected for the number of parameters (i.e.,*Ulrich, Ollik & Ugland, 2010*; *Alroy, 2015*)

Model fitting, log-likelihood, and AICc calculations were performed using Python (*Van Rossum & Drake, 2011*) and R (*R Core Team, 2016*). Python packages used for analysis include numpy (*Oliphant, 2007*; *Van der Walt, Colbert & Varoquaux, 2011*), matplotlib (*Hunter, 2007*), sqlalchemy (*Bayer, 2014*), pandas (*McKinney, 2010*), macroecotools (*Xiao et al., 2016*), and retriever (*Morris & White, 2013*). R packages used for analysis include ggplot2 (*Wickham, 2009*), magrittr(*Bache & Wickham, 2014*), tidyr (*Wickham, 2016*), and dplyr (*Wickham & Francois, 2016*). All of the code and all of the publicly available data necessary to replicate these analyses is available at https://github.com/weecology/sad-comparison and archived on Zenodo (*Baldridge et al., 2016*). The CBC datasets and NABA datasets are not publicly available and therefore are not included.

## RESULTS

Across all datasets, the negative binomial and Poisson lognormal distributions had very similar average log-likelihoods (within 0.01 of one another; Fig. 2). The log-likelihoods for each of these distributions averaged 0.8 units higher than for the log-series distribution and 5 units higher than for the Zipf distribution (corresponding to likelihoods that were twice as high and 140 times as high, respectively).

Although the negative binomial and Poisson lognormal distributions matched the data most closely, the likelihood provides a biased estimate of these distributions' ability to generalize to unobserved species. AICc approximately removes this bias by penalizing models with more degrees of freedom (e.g., the negative binomial and Poisson lognormal distributions, which have two free parameters instead of one like the log-series and Zipf distributions). After applying this penalty, the log-series distribution would be expected to make the best predictions for 69.2% of the sites. The Poisson lognormal and negative binomial distributions were each preferred in about 12% of the sites, and the Zipf distribution was preferred least often (6.0% of sites; Fig. 3).

Across all datasets and taxonomic groups, the log-series distribution had the highest AICc weights more often than any other model. The negative binomial performed well for BBS, but was almost never the best fitting model for plants (FIA and Gentry), butterflies (NABA), Acintopterygii, or Coleoptera. The Poisson lognormal performed well for the bird datasets (BBS and CBC) and the Gentry tree data, but was almost never best in the FIA and Coleoptera datasets (Fig. 4). The Zipf distribution only performed consistently well for Arachnida. Because datasets differ in both taxonomic groups and sampling methods care should be taken in interpreting these differences.

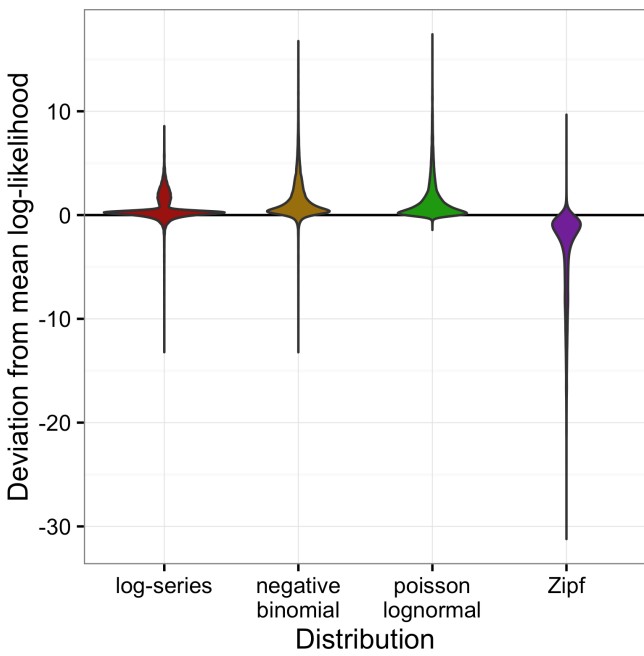

**Figure 2 Violin plots of the deviation from the mean log-likelihood for each site for all datasets combined.** Positive values indicate that the model fits better than the average fit across the four models.

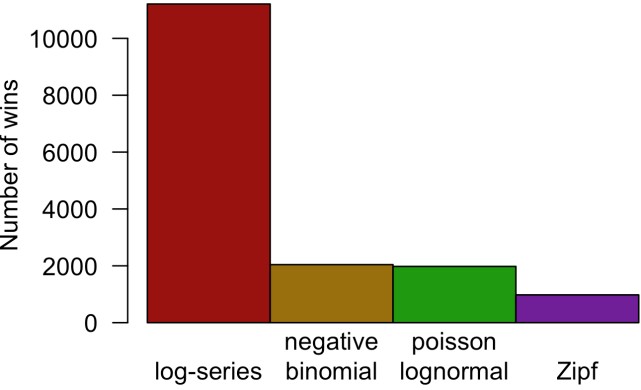

**Figure 3 Number of cases in which each model provided the best fit to the data based on AICc for all datasets combined.**

The full distribution of AICc weights shows separation among models (Fig. 5). Although the log-series distribution had the best AICc score much more often than the other models, its lead was never decisive: across all 16,209 sites, it never had more than about 75% of the AICc weight (Fig. 5). Most of the remaining weight was assigned to the negative binomial and Poisson lognormal distributions (each of which usually had at least 12–15% of the weight but was occasionally favored very strongly). The Zipf distribution showed a strong mode near zero, and usually had less than 7% of the weight.

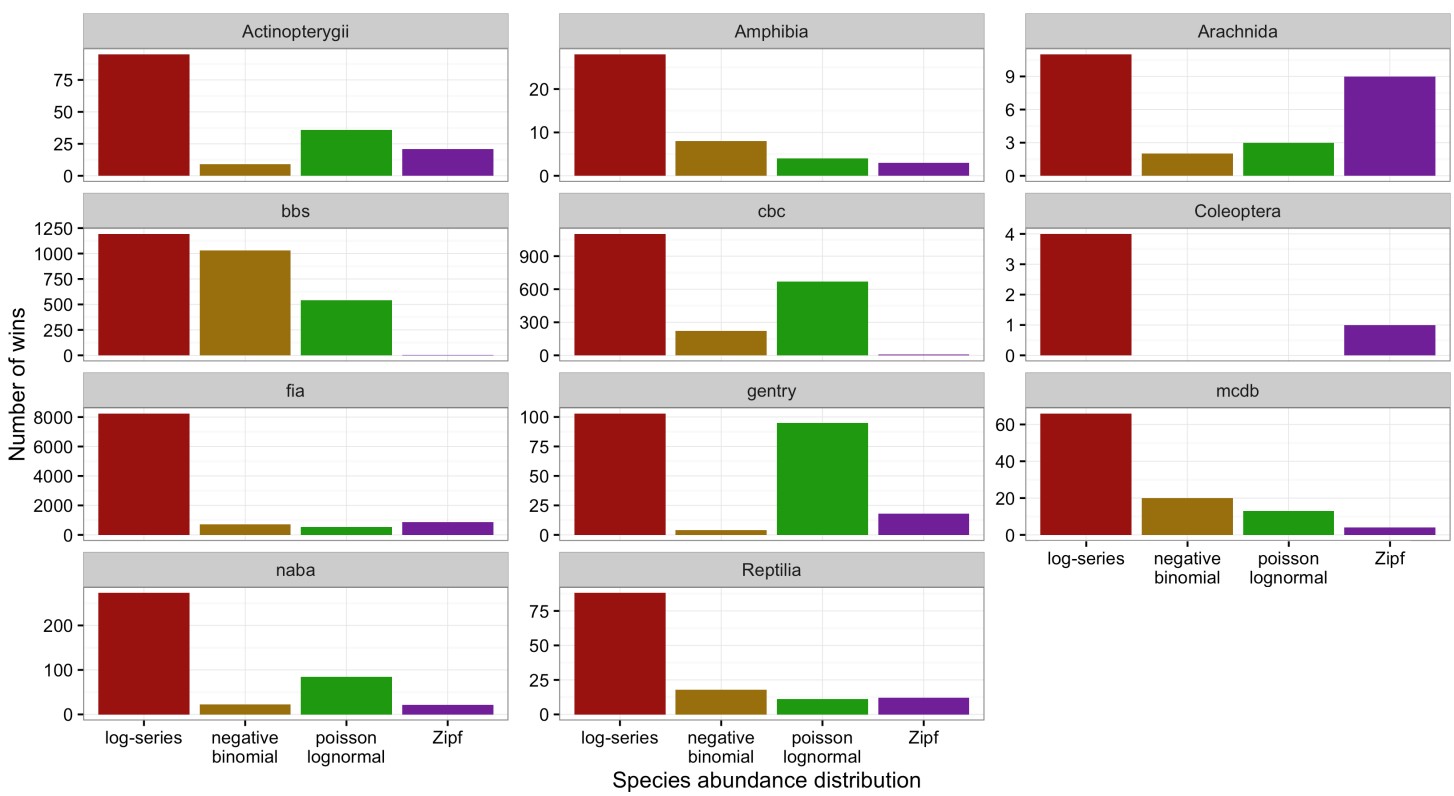

**Figure 4** Number of cases in which each model provided the best fit to the data based on AICc for each dataset separately.

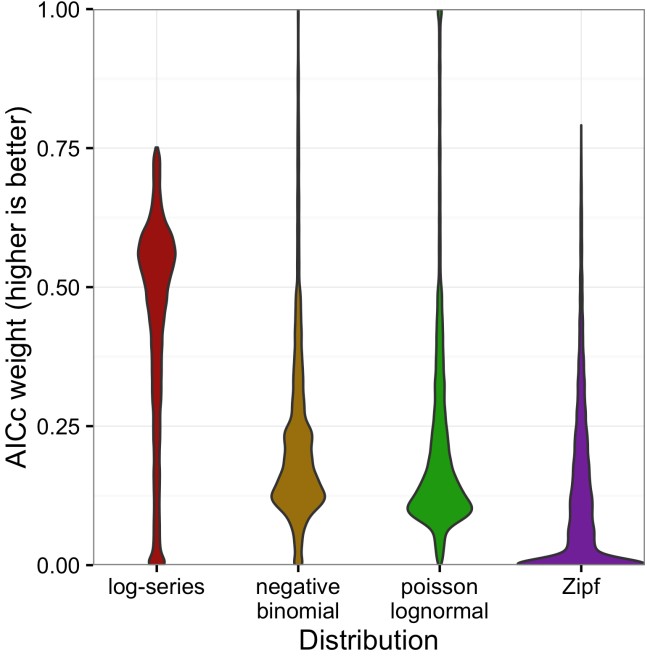

**Figure 5** Violin plots of the AICc weights for each model. Weights indicate the probability that the model is the best model for the data.

## DISCUSSION

Our extensive comparison of different models for the species abundance distribution (SAD) using rigorous statistical methods demonstrates that several of the most popular existing models provide equivalently good absolute fits to empirical data. Log-series, negative binomial, and Poisson lognormal all had model relative likelihoods between 0.25 and 0.5 suggesting that the three distributions provide roughly equivalent fits in most cases, but with the two-parameter model performing slightly better on average. Because the log-series has only a single parameter but fits the data almost as well as the two-parameter models, the log-series performed better in AICc-based model selection, which penalizes model complexity. These results differ from two other recent analyses of large numbers of species abundance distributions (*Ulrich, Ollik & Ugland, 2010*; *Connolly et al., 2014*) and are generally consistent with a third recent analysis (*Alroy, 2015*).

*Ulrich, Ollik & Ugland (2010)* analyzed ∼500 SADs and found support for three major forms of the SAD that changed depending on whether the community had been fully censused or not. They found that "fully censused" communities were best fit by the lognormal, and "incompletely sampled" communities were best fit by the Zipf and log-series (*Ulrich, Ollik & Ugland, 2010*). In contrast we find effectively no support for the Zipf across ecosystems and taxonomic groups, including a number of datasets that are incompletely sampled. Our AICc value results also do not support the conclusion that the lognormal outperforms the log-series in fully censused communities. The Gentry and FIA forest inventories both involve large stationary organisms and were collected with the goal of including all trees above a certain stem diameter. Therefore, above the minimum stem diameter, they are as close to fully censused communities as is typically possible. In these communities the log-series provides the best fit to the data most frequently. The discrepancy between our results and those found in (*Ulrich, Ollik & Ugland, 2010*) may be due to: (1) their use of binning and fitting curves to rank abundance plots, which deviates from the likelihood-based best practices (*Matthews & Whittaker, 2014*) used in this paper; (2) the statistical methods they use to identify communities as "fully censused", which tend to exclude communities with large numbers of singletons that would be better fit by distributions like the log-series; (3) the use of the continuous lognormal instead of the Poisson lognormal; (4) the fact that our censused communities are also a different taxonomic group from our sampled communities, making it difficult to distinguish between taxonomic and sampling differences.

*Connolly et al. (2014)* use likelihood-based methods to compare the negative binomial distribution (which they call the Poisson gamma) to the Poisson lognormal for a large number of marine communities. They found that the Poisson lognormal provides a substantially better fit than the negative binomial to empirical data and that the negative-binomial provides a better fit to communities simulated using neutral models. They conclude that these analyses of the SAD demonstrate that marine communities are structured by non-neutral processes. Our analysis differs from that in *Connolly et al. (2014)* in that they aggregate communities at larger spatial scales than those sampled and find the strongest results at large spatial scales. This may explain the difference between the two

analyses or there may be differences between the terrestrial systems analyzed here and the marine systems analyzed by *Connolly et al. (2014)*. The explanation for these differences is being explored elsewhere (SR Connolly et al., 2016, unpublished data).

*Alroy (2015)* compared the fits of the lognormal, log-series, Zipf, geometric series, broken stick, and a new model dubbed the "double geometric", to over 1,000 terrestrial community datasets assembled from the literature. To incorporate the geometric series, broken stick, and the double geometric, this research used non-standard methods for evaluating the fits of the models to the data, however the results were generally consistent with those presented here. The central Kullback–Leibler divergence statistics results showed that: (1) the Zipf, geometric series, and broken stick all perform consistently worse than the other distributions; (2) the double geometric, log-series, and lognormal all provide the best overall fit for at least one taxonomic group; and (3) the lognormal and double geometric fit the data equivalently well and slightly better than the log-series when not controlling for differences in the number of parameters (Alroy's Table S1, S2, and S3). Penalizing the two-parameter models (lognormal and double geometric) for their complexity, as we do here with AICc, would likewise improve the relative performance of the log-series distribution.

In combination, the results of these three papers suggest that in general the Zipf is a poor characterization of species-abundance distributions and that both the log-series and lognormal distributions provide reasonable fits in many cases. Differences in the performance of the log-series, lognormal, double geometric, and negative binomial, appear to be more minor. How these differences relate to differences in intensity of sampling, spatial scale, taxonomy, and ecosystem type (marine vs. terrestrial) remain open questions. Our analyses suggest that controlling for the number of parameters makes the log-series a slightly better fitting model, at least in the terrestrial systems we studied. Neither of the other papers that include the log-series (*Ulrich, Ollik & Ugland, 2010*; *Alroy, 2015*) make this correction and both show that it is still a reasonably competitive model even against those with more parameters.

The relatively similar fit of several commonly used distributions emphasizes the challenge of inferring the processes operating in ecological systems from the form of the abundance distribution. It is already well established that models based on different processes can yield equivalent models of the SAD, i.e., they predict distributions of exactly the same form (*Cohen, 1968*; *Boswell & Patil, 1971*; *Pielou, 1975*; *McGill et al., 2007*). To the extent that SADs are determined by random statistical processes, one might expect the observed distributions to be compatible with a wide variety of different process-based and process-free models (*Frank, 2009*; *Frank, 2011*; *Locey & White, 2013*). Regardless of the underlying reason that the models performed similarly, our results indicate that the SAD usually does not contain sufficient information to distinguish among the possible statistical processes—let alone biological processes—with any degree of certainty (*Volkov et al., 2005*), though it is possible that this result differs in marine systems (see *Connolly et al., 2014*). A more promising way to draw inferences about ecological processes is to evaluate each model's ability to simultaneously explain multiple macroecological patterns, rather than relying on a single pattern like the SAD (*McGill, 2003*; *McGill, Maurer & Weiser,*

*2006*; *Newman et al., 2014*; *Xiao, McGlinn & White, 2015*). It has also been suggested that examining second-order effects, such as the scale-dependence of macroecological patterns (*Blonder et al., 2014*) or how the parameters of the distribution change across gradients (*Mac Nally et al., 2014*), can provide better inference about process from these kinds of pattern.

## ACKNOWLEDGEMENTS

We thank all of the individuals involved in the collection and provision of the data used in this paper, including the citizen scientists who collect the BBS, CBC, and NABC data, the USGS and CWS scientists and managers, the Audubon Society, the North American Butterfly Association, the USDA Forest Service, the Missouri Botanical Garden, and Alwyn H. Gentry. We also thank all of the scientists who published their raw data allowing it to be combined in *Baldridge (2013)*.

### Funding

This research was supported by the National Science Foundation through a CAREER Grant 0953694 to Ethan White, and by the Gordon and Betty Moore Foundation's Data-Driven Discovery Initiative through Grant GBMF4563 to Ethan White. The funders had no role in study design, data collection and analysis, decision to publish, or preparation of the manuscript.

### Grant Disclosures

The following grant information was disclosed by the authors:
National Science Foundation: 0953694.
Gordon and Betty Moore Foundation's Data-Driven Discovery Initiative: GBMF4563.

### Competing Interests

Ethan P. White is an Academic Editor for PeerJ.

### Author Contributions

- Elita Baldridge conceived and designed the experiments, performed the experiments, analyzed the data, wrote the paper, prepared figures and/or tables, reviewed drafts of the paper.
- David J. Harris analyzed the data, contributed reagents/materials/analysis tools, wrote the paper, prepared figures and/or tables, reviewed drafts of the paper.
- Xiao Xiao performed the experiments, analyzed the data, contributed reagents/materials/analysis tools, wrote the paper, prepared figures and/or tables, reviewed drafts of the paper.
- Ethan P. White conceived and designed the experiments, performed the experiments, analyzed the data, contributed reagents/materials/analysis tools, wrote the paper, prepared figures and/or tables, reviewed drafts of the paper.

## Data Availability

Zenodo: https://doi.org/10.5281/zenodo.166725.

GitHub: https://github.com/weecology/sad-comparison.

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
