# Peer review of "An extensive comparison of species-abundance distribution models"

_PeerJ, doi:10.7717/peerj.2823_

## Round 0.1 · original submission · Minor Revisions

Dear Ethan,

Congratulations for your piece of work. It was a pleasure to read it.

Both reviewers did a nice work and they give you interesting feedback in order to improve the manuscript. Particularly, Reviewer 1 raised some concerns that you might solve by explaining with more detail your data and your methods.

all the best,
Sara Varela

·

Basic reporting

The manuscript #12365 presents a comparison among the performance of species-abundance models using a wide range of data sets from multiple communities and taxonomic groups. The study sounds good, the analyses and figures relevant, but it needs of a carefully revision of the language and a more appropriated discussion of the findings. Please, see specific suggestions and comments.

Follow bellow some points about the language, but it is needed a more carefully read and revision.

L. 64: Authors show Table 1, but did not cite it across the text. Please, provide citation for table 1.

L. 101-102: the sentence “best practices” is duplicated in this phrase.

L. 103-104: the word “therefore” is also duplicated in this phrase.

Title of Fig. 3: “Number of cases IN WHICH each model…” instead of “where”;
Idem Fig. 4!

L. 184: word “the” is duplicated;

L. 187: “negative-binomial” instead of “negatvie-binomial”;

Experimental design

In introduction (L. 34-39), authors justify one of the basic flaw of other studies about SADs is that they “use only a small subset of available models”. Next (L. 40), the authors describe the study’s goal and say that compared the performance of four SADs models. If one key question of investigation relates with the fitness of models, why that authors did not considered ALL existing SAD models in their analyses? These four chosen models often present the best performances (by the discussion, it seems that not!)? In methods, the authors justify their choice (the models describe discrete distribution, matching the type of data), but these questions remain unclear.

In methods, authors start saying that compiled a wide array of data sets from multiple sources and for multiple taxonomic groups, but do not say which kind of data they obtained. Are these data species occurrence records, species richness, population abundance…? Which is the temporal range of data? This is not clear in the text.

In lines 62-63 and 67-68, authors say that abundances were computed using counts of individuals. However, besides omit the kind of sampled raw data set, it is not clear what are the sampling units. For example, have the authors split the study area in grid cells (please, describe cell resolution)? Or the sampling units were ecoregions…?

Validity of the findings

L. 121-126: instead of just describe the list of R-packages used to perform analyses, please, describe which analyses were performed using each R-package. If possible, please, describe also the functions from each package used to perform analyses. A detailed description is a general practice from R-modelers and would clear the description of your methods.

L.172-174: I do not agree with this sentence. The authors did not compare model fitting between “fully” and “incompletely” sampled communities. It seems that authors considered all communities in their analyses, regardless of sampling effort. Thus, the presence of “fully censused” communities in this study may have negatively affected the fitness of Zipf models. Instead comparing the results of this study with other studies (especially Ulrich et al. 2010), I invite the authors to discuss about the implications of quality of data sets on model fitting;

L. 209-210: the author did not analyze the double geometric model in this study, so it is not a reasonable option to conclude that its performance appears to present minor difference with log-series, lognormal and negative binomial. This is speculation not supported by analyses;

L. 214-216: this is an excellent conclusion from this study! It is supported by analyses performed here and yet considerate that found in literature. Please, develop all the discussion on this way and avoid unsupported speculations.

Reviewer 2 ·

Basic reporting

The paper is very 'self-contained' with clear, focused goals that are reported and discussed in context to previous work. The data is made open when possible. The english language is good, in some cases it could a shortened but the manuscript is already short so this is not a problem.

Label sizes of Figure 4 should be increased if the plot is intended to be this size in the final article.

Experimental design

The experimental design is transparent and simple, and well-described. The knowledge gap is clearly stated, although in this case it is more of an 'information gap' that is filled by comparing more data sets. The authors improve on previous work by gathering more data sets and testing a greater number of SAD models, which is good.

Validity of the findings

The statistical methods are appropriate, the results seem solid and are well presented.

However, since each dataset reports findings from one taxa, I don't know how certain it is to state that the SAD is different for different taxa. This might just be an artifact of the sampling method within each dataset which is correlated to taxa. The authors discuss that trees are fully sampled yet the selected SAD model is different for both of them (log-series for fia, log-series and poisson for gentry, figure 4, lines 176-177). Are there datasets with several taxa that could be analysed? It would be interesting to break data up into sub-taxa within each category to assess if there are some general trends in SAD among subtaxa within a single taxa. For example, separate birds into passerines, parrots, falcons, pigeons, loons, etc, and see if the SAD model varies by subtaxa. Or by geographical regions.

Additional comments

The article is clearly and concisely written. It set out clear goals and fulfilled them. As the authors state in the conclusion, there are clearly more detailed ways to analyse the SAD, with multiple macroecological patterns or second-order effects, but that was not their goal.

minor corrections:
line
162 two parameter-model performing
184 remove one the
186 that the
187 negative
189 demonstrate
192 two analyses
200 showed that
207 those papers?
217 emphasize the challenge

---

## Round 0.2 · accepted · Accept

Thank you very much for improving your manuscript and for answering all the questions raised by the reviewers.